



# The global distribution of the M1 ocean tide

Philip L. Woodworth[1]

[1]National Oceanography Centre, Joseph Proudman Building, 6 Brownlow Street, Liverpool L3 5DA, United Kingdom

*Correspondence to*: Philip L. Woodworth (plw@noc.ac.uk)

**Abstract.** The worldwide distribution of the small degree-3 M1 ocean tide is investigated using a quasi-global data set of over 800 tide gauge records and a global tide model. M1 is confirmed to have a geographical variation in the Atlantic consistent with the suggestion of Platzman and Cartwright that M1 is generated in the ocean as a consequence of the spatial and temporal overlap of M1 in the tidal potential and one (or at least a small number) of diurnal ocean normal modes. As a

consequence, it is particularly strong around the UK and on North Sea coasts. This analysis shows that their suggestion is also consistent to a great extent with the observed small amplitudes in the Pacific and Indian Oceans. However, there are differences at the regional and local level which require much further study via more sophisticated ocean tidal modelling. By contrast, the M1´ tide is shown to have a geographical distribution consistent with expectations from other degree-2 diurnal tides, apart from locations such as around the UK where tidal interactions introduce complications. As far as we know, this is

the first time that these small tidal constituents have been mapped on a global basis and, in particular, the first time that the ocean response to the degree-3 component of the tidal potential has been investigated globally.

## 1 Introduction

M1 is a small tidal constituent with a frequency of 1 cycle/lunar day. It arises from the degree-3 component of the tidal

potential, unlike larger constituents such as M2, the predominant semidiurnal tide in the ocean with a frequency double that of M1, which originate from the degree-2 component. Agnew (2007), Pugh and Woodworth (2014) and other texts can be consulted for explanations of why the tidal potential contains degree-2, degree-3 (and -4 etc.) components.

M1 was not identified unambiguously in tide gauge records until as late as 1968 when it was observed in data from Cuxhaven, Germany (Cartwright, 1975). Instead, what analysts referred to as 'M1' was more likely to be due to other tidal

lines within the 'M1 group', that have frequencies slightly different from M1 itself and which originate from the degree-2 component of the tidal potential. Cartwright (1975, 1976) used very long tide gauge records (several with 18 or more years of data) to demonstrate that the true degree-3 M1 was particularly large (amplitude ~1 cm) around the UK and on North Sea coasts. This was consistent with the suggestion of Platzman (published later in Platzman, 1984b) that M1 is forced by one (or a small number) of normal modes, especially one of period 25.7 hours which is particularly strong in the Atlantic rather than



in the other ocean basins. Amin (1982) used an additional seven long records from the west coast of Great Britain, confirming the large M1 in this region and showing that amplitudes increased going north. Some years later, Cartwright et al. (1988) used 13 long tide gauge records from the North and South Atlantic, demonstrating consistency with Platzman's suggestion over a wider area. However, the authors (who included the present author) did not take the obvious next step of
checking that M1 was indeed much smaller in other ocean basins, and they made the rather rash statement that "We are unlikely to get any further data for this esoteric spectral line".

The aim of the present paper is to make that long-overdue extension to other ocean basins, and to densify Cartwright's findings for the Atlantic, by using data from over 800 tide gauge records distributed around the world.

Figure 1 is copied from Cartwright et al. (1988). It shows the amplitudes and Greenwich phase lags for M1 at the 13
Atlantic locations, some of which were taken from the earlier papers (Cartwright 1975, 1976). The co-range and co-tidal lines indicate the amplitudes and phase lags that Platzman obtained in his synthesis of M1 from about 10 normal modes of the world ocean (Platzman, 1984b), explaining, qualitatively at least, the larger amplitudes in the NE Atlantic. As Ray (2001) explained, M1 in the tidal potential is distributed symmetrically north and south of the Equator, unlike the more familiar degree-2 diurnal potential which is anti-symmetric. The 25.7 hour Platzman mode, the most important in the
synthesis, is also largely symmetric. Consequently, the similarity of M1 in the potential with the normal mode, both spatially and temporally, leads to its preferential excitation in the Atlantic where the mode is relatively strong.

The co-range and co-tidal lines in Figure 1 were copied from a figure in Platzman (1984b) which covered the whole ocean; that figure showing his synthesis of M1 is reproduced here as Figure 2. One can see that, for the Platzman-Cartwright theory of generation of M1 to be accepted as largely correct, then it remains to be verified by, for example, observations of
low amplitudes in the central Indian and Pacific Oceans, larger amplitudes in the North Pacific, and phase differences along the American Pacific coast.

## 2   Tidal Details

2.1 M1 and M1′

The 'M1 group' refers to a set of tidal lines with frequencies within 1 or 2 cycles/year of the frequency of M1 itself (Cartwright, 1975).[1] Table 1 lists the lines in the M1 group taken from the tables of Cartwright and Tayler (1971) and Cartwright and Edden (1973). These are denoted (1) to (8), with (4) being M1 itself with a frequency of 1 cycle/lunar day, or

---

[1] A 'group' in tidal terminology is a set of lines in the tidal potential that have the same first two Doodson numbers, so that different groups within the same species are separated by 1 cycle/month (Pugh and Woodworth, 2014). Cartwright (1975) was, therefore, being careful to explain what he meant by the 'M1 group'.



half that of M2. It has two nodal sidebands (3 and 5) which have approximately equal amplitudes (see below). These three result from the degree-3 component of the potential. As regards the degree-2 terms, we can ignore (6) which is much smaller than the other four lines (1, 2, 7 and 8). The combination of these four is denoted in this paper as M1′. In principle, if one has many years (ideally 18.6 but a minimum of 9, Cartwright, 1975) of good tide gauge data, then M1 and the two largest lines

in M1′ (2 and 7) can be identified individually, as in the Cartwright papers and Amin (1982).[2] However, they are not separable if one has only, say, one year of data. Consequently, the energy in a tidal spectrum obtained from one year of data might have a peak around the M1 frequency, but it will not be immediately apparent whether it stems from M1 itself or from M1′.

For example, Figure 3(a) shows the diurnal section of a spectrum of one year of data from Honolulu, Hawaii. The

main degree-2 tidal constituents (especially O1 and K1) stand out clearly, as does a peak around M1. In fact, it will be shown below that this peak is almost entirely due to degree-2 M1′. A similar situation applies in Figure 3(b) for Atlantic City (US Atlantic coast). Figure 3(c) has a spectrum for Cuxhaven (German North Sea coast), where M1 was first identified, and Figure 3(d) for Newlyn (SW England). It will be seen (as in Cartwright, 1975) that most of the peaks in (c,d) are due to degree-3 M1. The important point to make is that M1 (or M1′, to be decided) can usually be identified in such spectra above

the non-tidal background as long as it has an amplitude of several millimetres or more.

### 2.2  Usual Tidal Analysis Procedure for M1

In the software used at the National Oceanography Centre (NOC, Bell et al., 1998), and probably at other centres, the tidal

analysis of a tide gauge record makes allowance for a constituent named M1 with a frequency of 1 cycle/lunar day. However, in spite of its name, an assumption is made that any energy at that frequency is entirely due to what we call M1′, being a combination of the four largest degree-2 terms (1, 2, 7 and 8) in Table 1, in the same proportion as in the tidal potential. That energy is represented as a harmonic amplitude ($H$) and phase lag ($G$) which are adjusted by 'nodal factors' $f$ and $u$ that are time-dependent functions of the longitude of the lunar ascending node ($N$) and the longitude of lunar perigee

($p$). The usual expression 'nodal factor' is clearly a misnomer when, as in this case of M1′, the factors vary over a combination of both nodal and perigean cycles. Doodson (1928) and Doodson and Warburg (1941) show that the nodal factors for M1′ can be calculated from:

---

[2] The analyses of Cartwright (1975, 1976), Cartwright et al. (1988) and Amin (1982) involved spectral analyses of long tide gauge records in which lines (2), (4) and (7) in Table 1 were separable. However, one must note that the notation used was different. Cartwright refers to the principal degree-2 term (i.e. line 7) as M1′, whereas Amin (1982) refers to line (2) as M1′ and line (7) as M1″. IHO (2006) denotes lines 2 and 7 as M1B and M1A respectively. This confusing situation is not helped by misprints at the top of page 278 of Cartwright (1975) in which M1 and M1′ are interchanged, and in the header of Table 4(b) of Amin (1982). As explained above, the present paper takes it lead from Doodson (1928) and Doodson and Warburg (1941) to denote the combination of the four largest degree-2 terms (1, 2, 7 and 8) as M1′.



$$f \cos u = 2 \cos(p) + 0.4 \cos(p - N)$$

[1a]

$$f \sin u = \sin(p) + 0.2 \sin(p - N)$$

[1b]

with a normalisation in $f$ that stems from much earlier work by Darwin (see Doodson, 1921, 1928). As a consequence, its average value is approximately 1.57 and not 1.0. It is important to keep this normalisation in mind when comparisons are made to other reported amplitudes.

These expressions for $f$ and $u$ are still used in the NOC software and are shown by the black lines in Figure 4. It

can be seen that both factors change rapidly, primarily over the perigean cycle. A more exact computation, using the amplitudes in Cartwright and Tayler (1971) of the four largest degree-2 terms, results in the red and green lines respectively. There is little difference between the black and coloured lines, so we have continued to use Doodson's expressions for $f$ and $u$ in Equation 1 throughout the present investigation.

**3 Data and Methods**

The method used for this study is different to that used in the three Cartwright papers. The latter involved the use of a small number of long tide gauge records (several with 18 or more years of data), from which spectral analysis allowed the separation of M1 from the two main degree-2 lines. That method is admittedly more rigorous than the one described below, if one has long continuous records with few gaps. However, the method presented here works well for many disparate

records insofar as the degree-3 M1 constituent is concerned. And in the case of degree-2 M1′, it provides findings that are consistent for most of the world with Doodson's assumption that M1′ is composed of the 4 largest degree-2 terms in proportions given in the tidal potential, and consequently with the nodal factors for M1′ as in Equation 1.

The method employs records from the Global Extreme Sea Level Analysis Version 2 (GESLA-2) data set (Woodworth et al., 2017). A tidal analysis was performed for each station-year of data that was at least 75% complete, using

a standard set of 63 constituents including M1. That provided estimates of amplitude ($H$) and phase lag ($G$) for that year, as explained in Section 2.1. Subsequently, records were required to have at least 9 near-complete years of data for 1920-onwards, resulting in 804 records for study distributed around the world coastline. These come from 536 stations, there being sometimes alternative station records from more than one provider. Only 10 of these records were rejected, including those from Wilmington, North Carolina at which tides are known to have been affected by river dredging (e.g. Ray, 2009) and

several which had anomalous phase lags of the main tidal constituents, probably due to tide gauge timing errors.





The next step is to un-correct the estimated $H$ and $G$ for the nodal adjustments of Equation 1 that had been made in the tidal software. In other words, we calculate $h = Hf$ and $\varphi = G - u$, where the instantaneous (mid-year) amplitude ($h$) and phase ($\varphi$) can be thought of as representing an M1 carrier wave, which at this stage could be due to either degree-3 M1 or degree-2 M1′, or a combination of the two. One can then parameterise $h$ and $\varphi$ obtained for each year as a combination of

the contributions from M1′, which is the only part of the computed M1 that genuinely has the nodal variations of Equation 1, and M1 plus its nodal sidebands (3, 4 and 5 in Table 1). This parameterisation takes the form:

$$hcos(\varphi) = H_2 f cos(G_2 - u) + H_3 F sin(G_3 - U)$$

[2a]

$$hsin(\varphi) = H_2 f sin(G_2 - u) - H_3 F cos(G_3 - U)$$

[2b]

where the nodal factors for M1 ($F$ and $U$) are approximately $F = 1.0 - 0.28\,cos(N)$ and $U = 0.0$. The derivation of $F$ and $U$ for M1 with its approximately equal nodal sidebands is similar to that for Mm in Appendix A of Woodworth and Hibbert (2018).

The mixture of sines and cosines in Equation 2(a,b), and their signs, originate from the way these constituents are parameterised in the harmonic expansion employed in the tidal analysis. The use of Equation 2 assumes that $f$ and $u$ in reality are close to their values in Equation 1 and that $F$ and $U$ also have their equilibrium form. A least-squares search is then made for choice of $H_2$ and $G_2$ (the amplitude and phase lag of M1′) and $H_3$ and $G_3$ (the amplitude and phase lag of M1) that best describes Equation 2(a,b) for all the years in the record. The fit is aided by $f$ and $u$ varying so rapidly from year to

year.

The method results in reasonable agreement between M1′ phase lag and those of Cartwright, as shown in Table 2. There is agreement to within ~20° except for Southend and possibly Dunkerque. However, one should note that Cartwright's M1′ was the largest degree-2 term only (i.e. line 7 in Table 1), whereas in the present analysis M1′ refers to the combination of four terms, so comparison will not be perfect. In addition, because of the Doodson normalisation of $f$ values mentioned

above, the amplitudes for M1′ from this analysis appear systematically lower than Cartwright's. Reasonable agreement can be seen in most cases at the ~1 mm level if that is taken into account (see further below).

On the other hand, the present method results in almost identical findings for M1 to those of Cartwright. Results are also close to those of Amin (1982) for seven stations on the west coast of Great Britain and of Ray (2001) for three European stations (Table 3).

The Supplementary Material provides a list of the M1′ and M1 values obtained at all locations in the present work. For example, with reference to Figure 3, M1′ at Honolulu can be seen from the list to have an amplitude of 4.8 mm whereas



that of M1 is only 0.8 mm. At Atlantic City, M1′ and M1 have amplitudes in a similar proportion to Honolulu. On the other hand, M1 dominates at Cuxhaven and Newlyn.

## 4 Results for M1′ and M1

### 4.1 Maps of M1′

The amplitudes and phase lags for M1′ are shown in Figures 5(a,b) respectively. Amplitudes are ~10 mm or more along the Pacific coasts of N America, SE Asia, China and Japan (but much lower on the Sea of Japan Sea side), on the north coast of Australia and in the NW Indian Ocean. Phase lags show spatial consistency where amplitudes are large e.g. North American Pacific and around Australia.

However, it can be difficult to arrive at conclusions from inspection of coloured dots on maps like this (especially the phase lags). Therefore, the following method is useful in demonstrating that Figure 5 is consistent with expectations for a degree-2 diurnal tide. Supplementary Figure 1(a) shows a modern global map for K1, the main degree-2 diurnal tide. A map of a constituent with a similar frequency (such as P1) will look much the same, although with smaller overall amplitudes and small differences in the patterns of co-range and co-tidal lines. Another diurnal tide such as O1, with a larger difference in frequency to that of K1, will have a map with much larger differences, owing to the response of the ocean to forcing being different at the two frequencies. However, as the frequency of M1′ is mid-way between those of K1 and O1, its map should look something like an average of the two.

This can be tested by plotting values for K1 and O1 in the complex plane (Figure 6a), with amplitudes normalised by their average amplitudes in the data set (alternatively their amplitudes in the tidal potential could be used). They will be at different points in the plane because of the difference in the local response to forcing. One would expect M1′ to be located at a point in the plane mid-way between them, as shown by the open circle in Figure 6(a). That point has an amplitude R. The mis-match between the measured M1′ (after similar normalisation) and the mid-way point has a length D. Consequently, D/R provides as assessment of the accuracy of measurement of M1′. Figure 6(b) shows a histogram of this ratio, selecting stations for which K1 amplitude is larger than 10 cm, thereby excluding stations located near to diurnal amphidromes. It suggests that in most cases M1′ has been estimated with an accuracy of about 10%, at least for large M1′, and it implies that the use of Equation 1 for the nodal factors of M1′ was largely correct at most places. The requirement of a minimum K1 amplitude inevitably means that this test is performed for the higher amplitude regions in Supplementary Figure 1(a). Supplementary Figure 2(a) shows a worldwide map of D/R showing that it is largely a simple reflection of areas of ocean with large and small diurnal amplitudes, although with especially large values on the NW European continental shelf.



Amin (1982, 1985) obtained interesting results from his analysis of long records from the west coast of Great Britain, finding line (2) in to be several times larger than (7), whereas one would have expected (2) to be about 3 times smaller than (7) based on the tidal potential (Table 1). These results would invalidate Doodson's combination of four terms into M1′ and the nodal factors of Equation 1, at least for this area of ocean. Amin's findings for these stations have recently

been confirmed by analysis of very long (~40 year) UK records by Richard Ray (private communication). He concluded that line (7), which is the largest contributor to M1′ in the potential, is suppressed in this area by a compound tide NO1, which has the same frequency as line (7) and results from the interaction between N2 and O1. Ray's analysis also presented evidence for a sizeable SO1 (interaction between S2 and O1), which lends credibility to the existence of NO1.

Anomalous M1′ in this area in the present analysis is demonstrated by Amin's west coast stations having D/R ratios

in the range 0.4-1.4, as do other UK stations with the exception of SW England (Supplementary Figure 2b), even though most stations in this area have decimetric amplitudes for K1 and O1 (Pingree and Griffiths, 1982). Neighbouring stations on the European coasts have smaller values of D/R. Therefore, M1′ in this region has a different character to M1′ in other areas of ocean that also have at least decimetric diurnals but which have smaller D/R ratios, as shown in Figure 6(b).

To some extent, this interpretation is supported by looking at the variance (V) of the residuals of Equation 2(a,b) i.e.

the sum of the squares of the differences between the left and right sides of Equations 2(a,b) for all N years of data, divided by N. Supplementary Figure 3(a) shows a histogram of V for all the records in the analysis and Supplementary Figure 3(b) provides a worldwide map. Its median value is 0.037 cm$^2$, corresponding to a standard deviation of 2 mm. However, around the UK there can be much larger values (Supplementary Figure 3c), especially in the shallow areas of the Irish Sea and southern North Sea. In these cases, V is more typically 0.1-0.2 cm$^2$, or standard deviation of 3-4 mm. A large V could result

from a poor quality record, or from the parameterisation of Equation 2 being inappropriate, or both. It is hard to decide which explanation applies to a particular record. However, the spatial consistency shown in Figure 5(c) suggests that the parameterisation for M1′ might be to blame in these large-V areas. Lower values of V are found in SW England, towards the open ocean.

Cartwright (1975, 1976) concentrated only on line (7), presumably because his expectations from the potential led

him to believe that it would also be the most important degree-2 term in UK waters. His results for M1′ are in reasonable agreement with those obtained here (Table 2), with the possible exceptions of Southend and Dunkerque, as mentioned above. It can be seen from Supplementary Figure 3(c) that stations with lower values of V correspond to better agreement in Table 2, and that higher values apply to the west coast of Great Britain area of Amin's stations and to the southern North Sea area of Southend and Dunkerque. Therefore, if Amin and Ray are correct that an interaction term (NO1) also plays a role at the

frequency of line (7) at some locations, then Cartwright's focus on (7) and our computation of M1′ in the larger-V areas will not be so meaningful. Nevertheless, the findings for M1′ for the UK overall (Supplementary Figure 4(a,b)) do appear similar





to those for other diurnals (K1 and O1), having somewhat lower amplitudes on the west coast of Great Britain than the east coast, and a clockwise rotation of phase lag around most of the coast (Pingree and Griffiths, 1982).

For completeness, Supplementary Figures 3(d,e) provide maps of V for NE and NW America respectively. Spatial consistency is again demonstrated for the largest values of V in the Maritime Provinces of Canada, Mid-Atlantic Bight and

between Vancouver Island and the mainland, all shallow-water areas where diurnal amplitudes are large and where interaction contributions to M1′ are plausible. Similarly there are large values along the coasts of China and SE Asia (Supplementary Figure 3b).

### 4.2 Maps of M1

Figure 7 (a,b) shows the corresponding maps for M1 amplitude and phase lag respectively, As shown by Cartwright (1975, 1976), Amin (1982) and Cartwright et al. (1988), there are large (~1 cm) amplitudes around the UK and along North Sea coastlines. There are moderate (several mm) amplitudes in NE Canada, as Cartwright (1975) suggested might be the case, but much smaller amplitudes to the south along most of the North American Atlantic coast, until one reaches the Gulf of

Mexico and Caribbean where moderate values are obtained again. Similarly, amplitudes of several mm are found on the American and African coasts of the South Atlantic. Figure 7(b) shows M1 phase lag, confirming at a large scale the rotations in the North and South Atlantic from Platzman's synthesis discussed by Cartwright et al. (1988). The co-tidal lines from the synthesis taken from Figure 1 are also shown; they help to guide the eye around the coloured dots on the coastlines.

Amplitudes of M1 are very small for most of the central Pacific. For example, as mentioned above, the M1

amplitude at Honolulu (Figure 3a) is only 0.8 mm compared to 4.8 mm for M1′. An exception is the coast of the Canadian Pacific and Alaska where amplitudes are ~3 mm as they are for southern Chile. Values of several mm are obtained around Australia, with larger values on the west and north coasts (although at least one of the red dots in north Australia in Figure 7(a) is anomalous, being from Wyndham which is up a river, while the red dot in South Australia is for Port Pirie in Spencer Gulf). Amplitudes are smaller on the east side of New Zealand than on its west coast. Moderate values (~4 mm) are found

for SE Asia and southern Africa, while there are higher amplitudes (~7 mm) in the NW Indian Ocean, where there are similarly high amplitudes for M1′. (A separate analysis for Karachi, Pakistan of a combination of two short records, which are individually shorter than the 9 years required in Section 3 and so do not appear in Figure 7, confirms a similarly high amplitude for M1 of ~6.2 mm in the NW Indian Ocean.) Amplitudes at only the 1-2 mm level are observed for several islands in the central Indian Ocean. Overall, these features in Figure 7(a) are qualitatively consistent with Platzman's

synthesis in Figure 2.

Figure 7(b) shows several interesting features for M1 phase lag, other than those already pointed out for the Atlantic by Cartwright et al. (1988), such as differences between the central part of the American Pacific coast and the northern and





southern sections of that coastline. Contrasts in M1 phase lag can be seen in Figure 7(b) between the east and west sides of New Zealand, Australia and Japan. There is reassuring consistency in both amplitude and phase information between the small number of stations in the NW Indian Ocean, and similarly between stations in the Indian Ocean sector of the Antarctic coast. The clockwise phase rotation in the Indian Ocean in Figure 2 might be supported by Figure 7(b) but there is no
evidence for a similar rotation in the central Pacific.

Additional remarks can be made on M1 from Supplementary Figures 5 (a-f) which focus on NW Europe, NE and NW America.

As regards the UK, an increase in the amplitude of M1 can be seen travelling north on the west side of Great Britain (Supplementary Figure 5(a)), confirming the results of Cartwright (1975) and Amin (1982). Amplitudes are mostly smaller
on the west coast of Great Britain than the east coast, although they are particularly large in the Bristol Channel (amplitudes of 24, 20 and 15 mm at Avonmouth, Newport and Hinkley Point respectively). Amplitudes are lower on the south coast of England than on the adjacent French coast, although the phases are similar. In addition, there is a phase difference of ~180° between the central-west and south-east coasts of Great Britain (Supplementary Figure 5(b)). Phase lag is smaller in the SW and NW than in the centre of the west coast, which is consistent with Amin (1982). He found it hard to decide on the spatial
variation of phase lag around the coast from only 7 stations, but the pattern is now quite clear from the many additional records. The Atlantic coast of France is 180° out of phase with the southern North Sea. Phase lag in NE Scotland is similar to that along the Norwegian coast.

Large amplitudes (~10 mm) are found at stations on the European coast of the North Sea, including Cuxhaven, with phase lag appearing to increase going north from the Netherlands to Norway. Amplitudes are lower (~5 mm) along the
Norwegian coastline. Further east, Voinov (2011) calculated amplitudes for M1 of 7 mm at two stations in the Barents Sea, and 2 and 5 mm at two stations in the Kara Sea. Supplementary Figure 5(a) shows amplitudes at the 1-2 mm level in the Baltic and Mediterranean, with the notable exception of Trieste, Italy where amplitudes were obtained in the present analysis of 4.6 mm for M1´ and 4.0 mm for M1. An M1 amplitude of 9.1 mm was reported at Trieste by Mosetti and Manca (1972) which was probably due to a combination of the two.

Supplementary Figure 5(c) demonstrates how moderate amplitudes (~4 mm) in NE Canada and in the Gulf of Mexico are separated by an extensive section of Atlantic coast where M1 amplitudes are minimal. Two stations with larger M1 amplitudes are located in the Delaware estuary where local tidal processes are probably the cause. Even though most amplitudes in this region are very small, the phase lags obtained for most stations are consistent with those of their neighbours, providing confidence in the findings (Supplementary Figure 5(d)). Two exceptions are located in the
Chesapeake estuary where again local processes are probably affecting the tides (Ross et al., 2017).

Amplitudes on the Pacific coast of NW America increase as one travels north of San Francisco (Figure 7a), with values of typically 3-4 mm on the Canadian and Alaskan coast (Supplementary Figure 5(e)), and larger values of 4-6 mm



between Vancouver Island and the mainland where enhancements of the degree-2 diurnal tides are also found (Foreman et al., 2000). Phase lags are consistent from station to station with an exception of Anchorage at the head of the Cook Inlet (Supplementary Figure 5(f)).

### 4.3 A numerical model for M1

The Platzman normal modes were computed many years ago on a coarse grid, with the M1 synthesis constructed from a combination of a small number of non-dissipative modes, together with a variational treatment to accommodate dissipation (Platzman, 1984a,b). Instead of following that approach in the present work, it was decided to construct a numerical model for M1 in which the modes would, in principle, be included implicitly.

The model is a global version of the regional tide-surge model of Roger Flather (e.g. Flather, 1988). The present version uses a ¼-degree bathymetry derived from the General Bathymetric Chart of the Oceans (GEBCO) (Weatherall et al., 2015). It must be stressed that the model can in no way be compared to the several state-of-the-art global tide models now available, most of which make use of information from satellite altimetry (Stammer et al., 2014). Instead, it should be considered as a tool for the present purpose of investigating the spatial variation of M1. The Supplementary Material

provides details of model construction.

A first step was to obtain an optimum tuning of the model for different selections of bottom friction and horizontal eddy viscosity, each time driven by the tidal potential for the four main degree-2 constituents only (M2, S2, K1 and O1). When acceptable global maps for M2 and K1 were obtained, then the model was considered tuned. (The Supplementary Material contains more details on how well M2 and K1 are simulated.) A second step was to run the model using only M2,

S2 and M1. In this case, K1 and O1 were left out of the forcing in order to avoid any leakage into M1 during a short (14 days) model run. In this exercise for M1, the driving potential is not proportional to $sin(2\varphi)$ as for degree-2 diurnals (where $\varphi$ is latitude) but has a value of $0.0012\ cos\ (\varphi)[5\ sin^2(\varphi) - 1]$ metres (Cartwright and Tayler, 1971). This is a tiny amplitude, even before multiplication by a degree-3 diminishing factor of 0.80 (Wahr, 1991). Nevertheless, it will be seen to be capable of generating an M1 of the order of a centimetre in parts of the ocean, such as the NE Atlantic, as discussed

above.

Figure 8(a,b) shows the resulting model amplitudes and phase lags for M1 using the same colour scales as for the tide gauge data in Figure 7(a,b). Figure 8(a) indicates larger amplitudes in the NE Atlantic, NW Australia and the NW Indian Ocean that are largely supported by the tide gauge data. Moderate amplitudes are also found in NE Canada, Gulf of Mexico, the South Atlantic coasts of America and Africa, the NE Pacific American coast, and along the Antarctic coastline. The US

Atlantic coast has very low amplitudes, as do the extensive central areas of the Pacific and Indian Oceans.

Model phase lags are shown in Figure 8(b) as a colour grid to enable easier comparison to those obtained from tide gauges in Figure 7(b). The two main amphidromic centres in the North and South Atlantic can be seen, although at slightly





different positions than in Figure 1. In addition, there is a clear pattern of rotation in the central Indian Ocean, different phase lags on the west and east coasts of Australia, and even similar phases in model and tide gauge data across the Pacific. The variation of phase lag along the Pacific coast of the Americas is similar to that in the tide gauges, except near to the Equator where the phase varies rapidly. Overall, there can be seen to be reasonable agreement between model and data.

5        Many aspects of the spatial variation of M1 described by the model and the tide gauges are also to be seen in the Platzman synthesis of Figure 2. Consequently, these findings can be taken as confirmation of the Platzman-Cartwright suggestion for the generation of M1. A possible exception concerns the pattern of amphidromes in the Pacific and the progression of phase lag along the American Pacific coast which differ somewhat from Platzman's synthesis. However, this is probably acceptable given the low amplitudes in this region.

## 5 Conclusions

The method for determining M1´ and M1 amplitudes and phases via Equation 2(a,b), and the availability of data from the large number of tide gauge records in the GESLA-2 data set, has enabled these small constituents to be studied on a near-global basis for the first time. Even though M1´ and M1 may have small amplitudes, and at first sight it seems remarkable

that they can be measured reliably at all, the general consistency of findings between neighbouring stations provides confidence in the results.

        The worldwide distribution of M1´ has been shown to be consistent with expectations for a degree-2 diurnal tide, being similar in its general character to other diurnals (K1 and O1). As a result, Doodson's method of combination of four degree-2 terms into an overall M1´ seems justified. However, there is some uncertainty regarding this way of computing M1´

at stations around the UK and on the NW European continental coastline where interaction between N2 and O1 into NO1, at the same frequency as line (7), could provide a complication. It seems that to do a better job for M1´ around the UK, one must abandon the Doodson assumption of nodal factors in Equation 1, and return to the analysis of long, near-complete records of (usually) hourly data, as Amin (1982) demonstrated.

        A much better situation applies for M1, with excellent agreement between the results of methods used in previous

and present studies (Tables 2 and 3). This is the first time that this esoteric degree-3 ocean tide has been studied in any detail beyond NW Europe, where it was first identified, and the coastlines of the Atlantic Ocean. Inspection of tide gauge findings in Figure 7(a,b), and of the spatial variations in M1 amplitude and phase from the numerical model in Figure 8(a,b), indicates that the Platzman-Cartwright suggestion was correct. The ocean responds to degree-3 forcing at M1 frequency as a result of normal modes with similar frequencies, consistent with Cartwright's findings from the Atlantic and also now from the other

ocean basins. In particular, the small amplitudes over most of the central Pacific and Indian Oceans, the larger amplitudes along the NE Pacific coast, and the apparent rotation of phase lag in the Indian Ocean are all qualitatively in agreement with the Platzman synthesis in Figure 2.



However, there are also detailed differences on a regional scale, for example the difference phases observed in the tide gauge data either side of New Zealand and Japan, which remain for further study. And on a local scale, there are anomalies in estuaries and inlets where shallow-water processes might result in apparent M1 generation although it is not immediately clear how (interaction of N2 and O1, for example, would generate an apparent M1´ rather than M1). Such non-linear processes for M1 generation on a larger scale were considered and rejected by Cartwright (1975).

Consequently, while the maps of Figure 7(a,b) and in the Supplementary Material are interesting, and their qualitative agreement with the numerical model and with Platzman's synthesis is reassuring, there is clearly a need for more detailed studies using more advanced ocean tide modelling. To our knowledge, a global model for M1 has never been constructed until now. For example, M1 is not included in the Finite Element Solution 2014 model (FES, 2018) that is now used by many groups. Such studies would provide further insight into how the ocean responds to a different (degree-3) form of tidal forcing and would be an important extension of the pioneering work on normal modes of the world ocean by Platzman.

*Acknowledgements.* I would not have had the idea for this paper without the previous work by David Cartwright and colleagues at Bidston Observatory. I am grateful to Cartwright in particular, to Roger Flather for the model code, and to David Pugh, Ian Vassie, Richard Ray and others for discussions about tides. Ray is also thanked for sharing his analyses of UK records. Some figures in this paper were generated using the Generic Mapping Tools (Wessel and Smith, 1998).

*Data availability.* All tide gauge data used in this paper may be obtained from https://www.gesla.org.

*Competing interests.* The author declares that he has no conflict of interest.

*Special issue statement.* This article is part of the special issue "Developments in the science and history of tides (OS/ACP/HGSS/NPG/SE inter-journal SI)". It is not associated with a conference.

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





**Table 1.** Lines in the tidal potential within the M1 group.

| Line | Doodson Number | | | | | | Degree | Frequency (cycle/hr) | Amplitude(*) |
|------|---|---|---|---|---|---|--------|-----------|--------------|
| (1) | 1 | 0 | 0 | -1 | -1 | 0 | 2 | 14.48520 | 0.00137 |
| (2) | 1 | 0 | 0 | -1 | 0 | 0 | 2 | 14.48741 | 0.00741 |
| (3) | 1 | 0 | 0 | 0 | -1 | 0 | 3 | 14.48985 | 0.00059 |
| (4) | 1 | 0 | 0 | 0 | 0 | 0 | 3 | 14.49205 | 0.00399 |
| (5) | 1 | 0 | 0 | 0 | 1 | 0 | 3 | 14.49426 | 0.00052 |
| (6) | 1 | 0 | 0 | 1 | -1 | 0 | 2 | 14.49449 | 0.00059 |
| (7) | 1 | 0 | 0 | 1 | 0 | 0 | 2 | 14.49669 | 0.02062 |
| (8) | 1 | 0 | 0 | 1 | 1 | 0 | 2 | 14.49890 | 0.00414 |

(*) Amplitudes are from the column in the tables of Cartwright and Tayler (1971) and Cartwright and Edden (1973) which refers to the mid-20[th] century with their signs omitted (the signs provide phase information). For degree-2 we consider only lines with amplitudes larger than 0.00015. For present purposes, the amplitudes can be considered as having arbitrary units. If an amplitude is required for a particular location then it has to be multiplied by its spatial dependence given in Table 2 of Cartwright and Tayler (1971) to provide a value in metres. Note that line (4) in this list is the real M1.





**Table 2.** Amplitudes (mm) and Greenwich phase lags (deg) for M1´ and M1 as reported in Cartwright (1975, 1976) and Cartwright et al. (1988). Figure 5 of the latter shows M1 values for 13 stations of which 6 had been reported in the earlier papers. In the right-hand columns are the corresponding values obtained in the present analysis (+).

| | Cartwright Papers | | | | Present Analysis | | | | |
| | M1´ | | M1 | | M1´ | | M1 | | |
| | $H_2$ | $G_2$ | $H_3$ | $G_3$ | $H_2$ | $G_2$ | $H_3$ | $G_3$ | |
|---|---|---|---|---|---|---|---|---|---|
| Cartwright (1975) | | | | | | | | | |
| Newlyn, Cornwall (*) | 1.2 | 38 | 4.8 | 278 | 1.2 | 32 | 4.6 | 274 | |
| Scilly Isles | 3.1 | 43 | 5.0 | 276 | 1.8 | 24 | 4.6 | 272 | |
| Brest, Britanny | 3.0 | 31 | 6.9 | 268 | 2.0 | 28 | 7.0 | 262 | |
| Malin Head, Eire | 5.6 | 101 | 8.9 | 312 | 2.0 | 72 | 9.0 | 320 | |
| Stornoway, Hebrides | 6.7 | 81 | 8.8 | 308 | 2.4 | 62 | 10.0 | 296 | |
| Lerwick, Shetland | 3.7 | 91 | 7.4 | 330 | 2.2 | 90 | 7.0 | 330 | |
| Southend, Essex | 2.5 | 88 | 9.5 | 112 | 2.0 | 174 | 8.4 | 116 | (++) |
| Dunkerque, France | 6.6 | 101 | 9.2 | 104 | 2.0 | 154 | 8.0 | 98 | |
| Cuxhaven, Germany | (**) | | 9.2 | 214 | 1.6 | 282 | 9.2 | 212 | |
| Terceira, Azores | 2.2 | 41 | 2.1 | 275 | 1.4 | 22 | 2.8 | 252 | (+++) |
| St. Georges, Bermuda (***) | 3.4 | 191 | 1.0 | 84 | 2.6 | 188 | 0.8 | 124 | |
| Cartwright (1976) | | | | | | | | | |
| Lagos, Portugal | 4 | 5 | 5 | 245 | 2.6 | 358 | 4.4 | 242 | |
| Cartwright et al. (1988) | | | | | | | | | |
| Reykjavik, Iceland | | | 6 | 308 | | | 4.2 | 310 | |
| St. John's, Newfoundland | | | 2 | 8 | | | 3.6 | 4 | |
| Atlantic City, USA | | | 2 | 100 | | | 0.6 | 82 | |
| Cananeia, Brazil | | | 5 | 124 | | | 5.0 | 124 | |
| Port Nolloth, S. Africa | | | 4 | 236 | | | 3.6 | 228 | |
| Simons Bay, S. Africa | | | 4 | 250 | | | 4.0 | 244 | |
| Vernadsky, Antarctica | | | 3 | 33 | | | 3.4 | 38 | |





(*) Average of 3 separate Newlyn records reported in Cartwright (1975)

(**) Said to be below noise level.

(***) The Bermuda M1 phase lag is given as 260 degrees in Cartwright (1975) and 84 degrees in Cartwright et al. (1988). The latter seems to be the more reliable.

(+) Where there was more than one alternative record for a station in the GESLA-2 data set, the records from the British Oceanographic Data Centre were used in this table for comparison to the British stations in Cartwright's papers. Similarly, records from French, German, Icelandic and US

national agencies were used where available (REFMAR, BSH, Coastguard and NOAA respectively). Otherwise, records were used from the University of Hawaii Sea Level Center. See the Supplementary Data Set for a full list of M1' and M1 values from the present analysis. Note also that M1' in Cartwright (1975, 1976) refers to line (7) in Table 1 only, while in the present

analysis it refers to the combination of four lines (1, 2, 7, 8).

(++) Sheerness used in the present analysis.

(+++) Ponta Delgada used in the present analysis.





**Table 3.** Amplitudes (mm) and Greenwich phase lags (deg) for M1 as reported in Amin (1982) and Ray (2001) and the corresponding values in the present analysis.

|  | Amin (1982) | | Present Analysis | |
|---|---|---|---|---|
|  | $H_3$ | $G_3$ | $H_3$ | $G_3$ |
| Millport | 7.5 | 348 | 8.8 | 336 |
| Portpatrick | 7.0 | 2 | 7.8 | 348 |
| Heysham | 5.9 | 352 | 10.2 | 358 |
| Liverpool | 7.2 | 30 | 9.8 | 4 |
| Holyhead | 5.7 | 332 | 7.2 | 330 |
| Fishguard | 5.0 | 313 | 6.2 | 310 |
| Milford Haven | 5.4 | 299 | 5.8 | 288 |

|  | Ray (2001) | | Present Analysis | |
|---|---|---|---|---|
|  | $H_3$ | $G_3$ | $H_3$ | $G_3$ |
| Newlyn | 4.67 ± 0.22 | 275 ± 3 | 4.6 | 274 |
| Lerwick | 7.25 ± 0.18 | 332 ± 2 | 7.0 | 330 |
| Vigo | 5.48 ± 0.15 | 232 ± 2 | 5.2 | 230 (*) |

(*) Using the record from the University of Hawaii in the GESLA-2 data set for the present analysis, instead of the record from the Instituto Español de Oceanografia which gives an amplitude of 5.4 mm and phase lag of 248°.



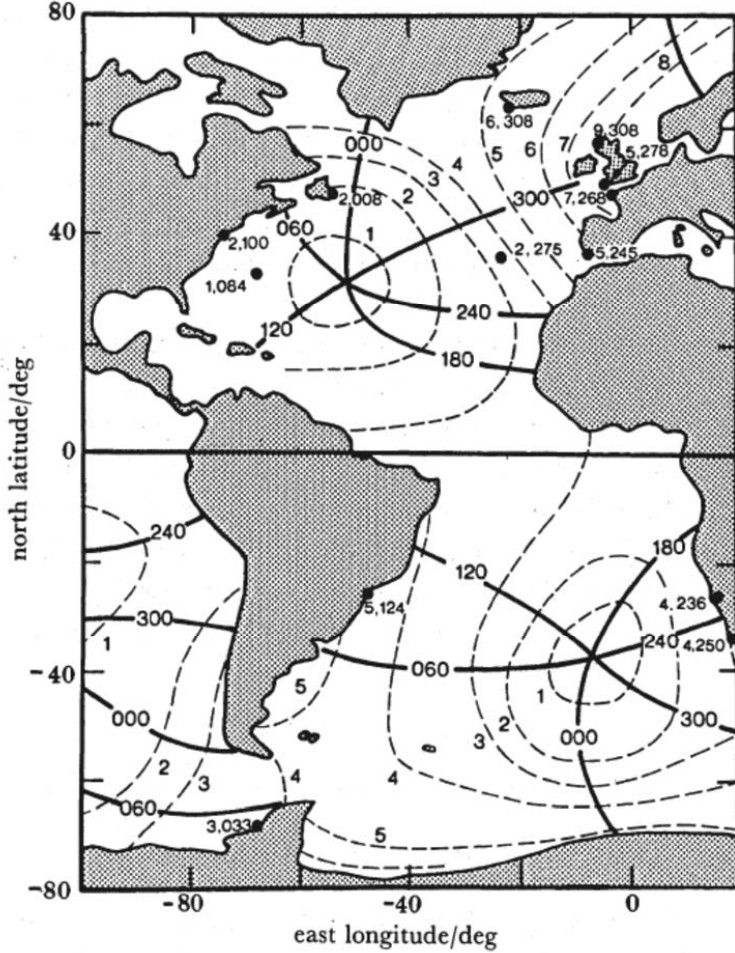

**Figure 1.** Contours of equal amplitude (millimetres, dashed lines), and equal phase lag (degrees, full lines) synthesized for the component M1, transposed from Figure 9 of Platzman (1984b) with 60° arbitrarily added to phase values. Black circles are positions of multiyear coastal stations at which M1 has been directly evaluated, with (amplitude, phase lag) shown alongside each station location. From Cartwright et al. (1988).





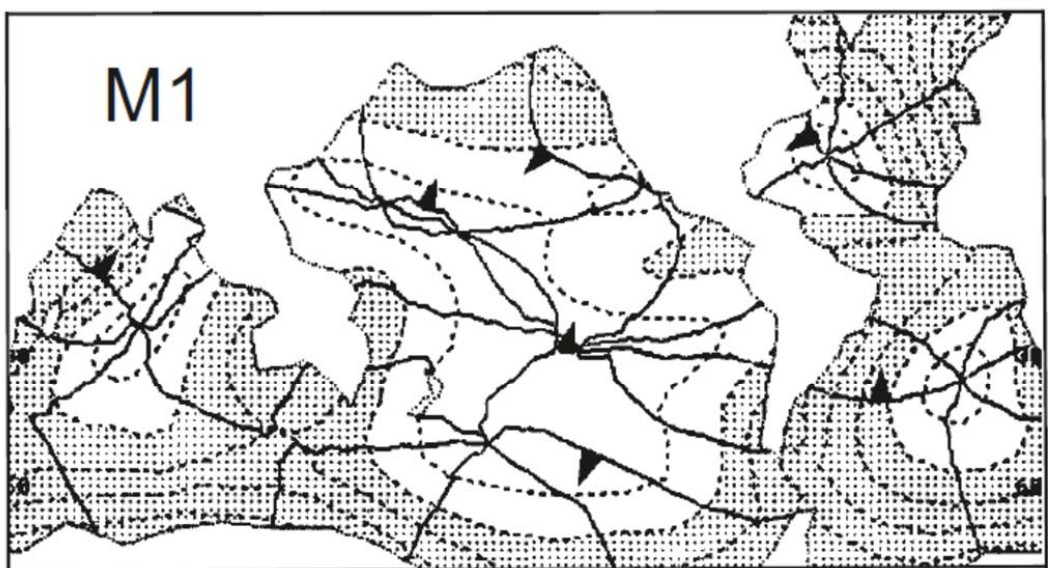

**Figure 2**. A map of the synthesized principal diurnal tide of third degree (M1). Amplitudes (dashed lines) are contoured at 1 mm intervals, and are shaded where the amplitude exceeds 2 mm. Phases are shown at 60° intervals, and the direction of phase propagation (increasing phase) is indicated by an arrowhead on contours of zero phase. This figure is copied from a
5  panel in Figure 9 of Platzman (1984b). Cartwright et al. (1988) had to add 60° to the phases shown here in order to represent adequately the tide gauge measurements of M1 phase, as shown for the Atlantic in Figure 1; the reasons for the offset were not understood.





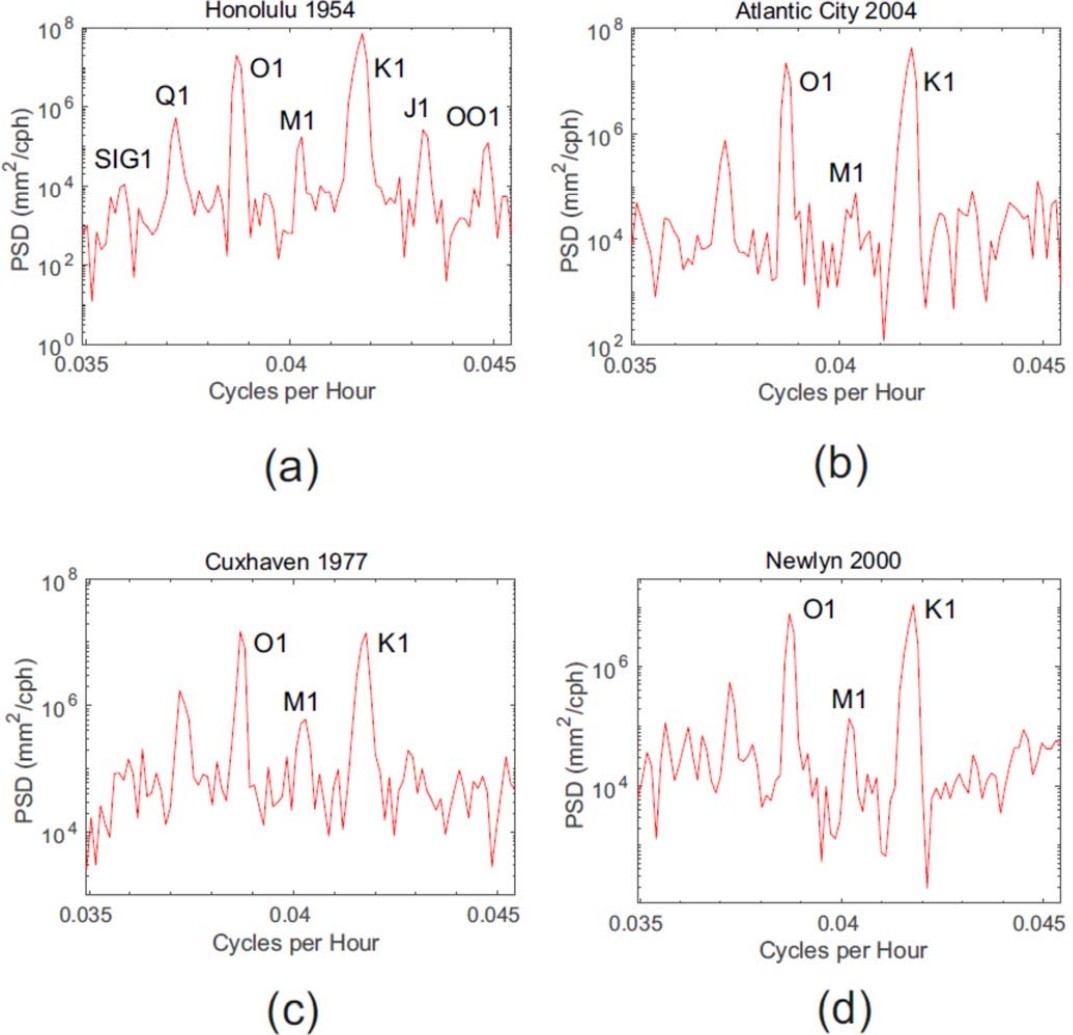

**Figure 3.** (a) Power spectral density (PSD) within the diurnal section of a spectrum of one year of sea level data from Honolulu, Hawaii showing the lines of the main diurnal tides. (b-d) The corresponding spectra from Atlantic City, Cuxhaven and Newlyn respectively.





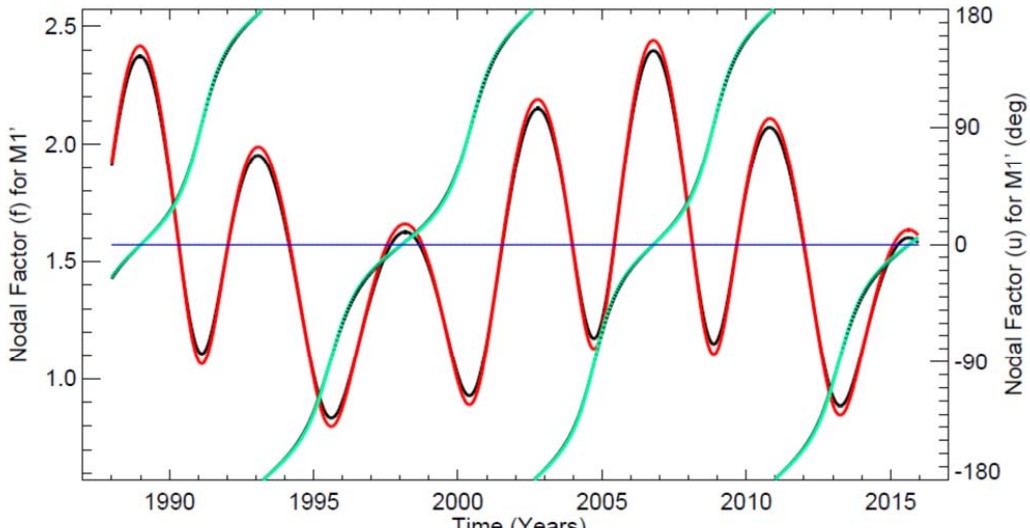

**Figure 4.** Variation of $f$ as given by Equation 1(a,b) as a function of time (in black), with an average value of approximately 1.57 owing to an historical normalisation explained in Doodson (1928). This is accompanied closely (in red) by values one would obtain using amplitudes in Cartwright and Tayler (1971), normalised so as to have the same average value. The $f$ values are with respect to the left axis. And variations in $u$ from Equation 1(a,b) (in black), closely accompanied (in green) by the corresponding values using Cartwright and Tayler (1971). The $u$ values are with respect to the right axis. The $f$ values in the second case span a slightly larger range than in the first case, while $u$ values in the two cases are almost identical and so the black and green curves overlap.



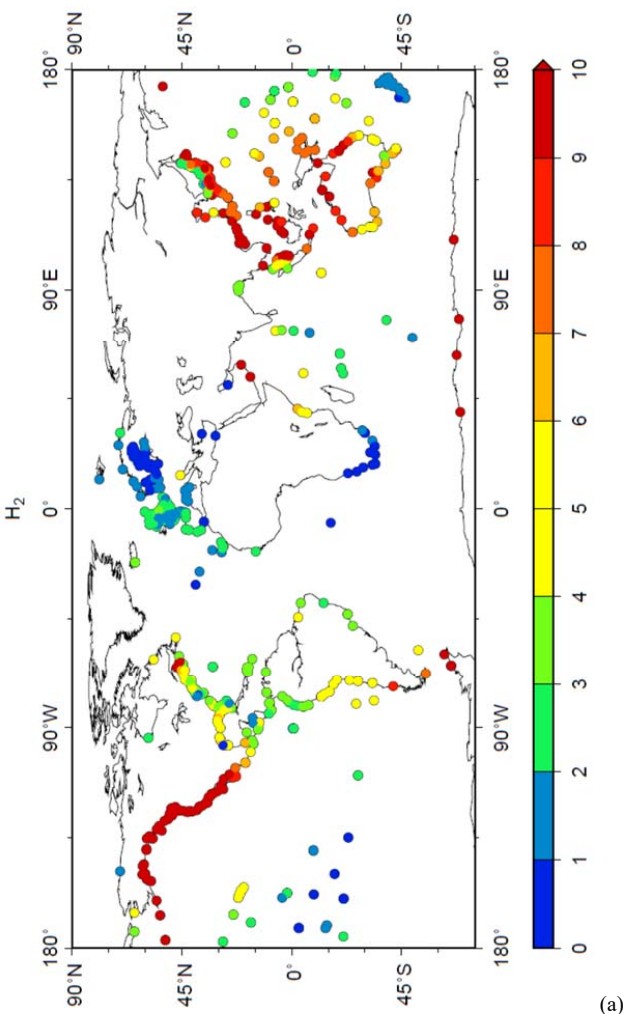

(a)





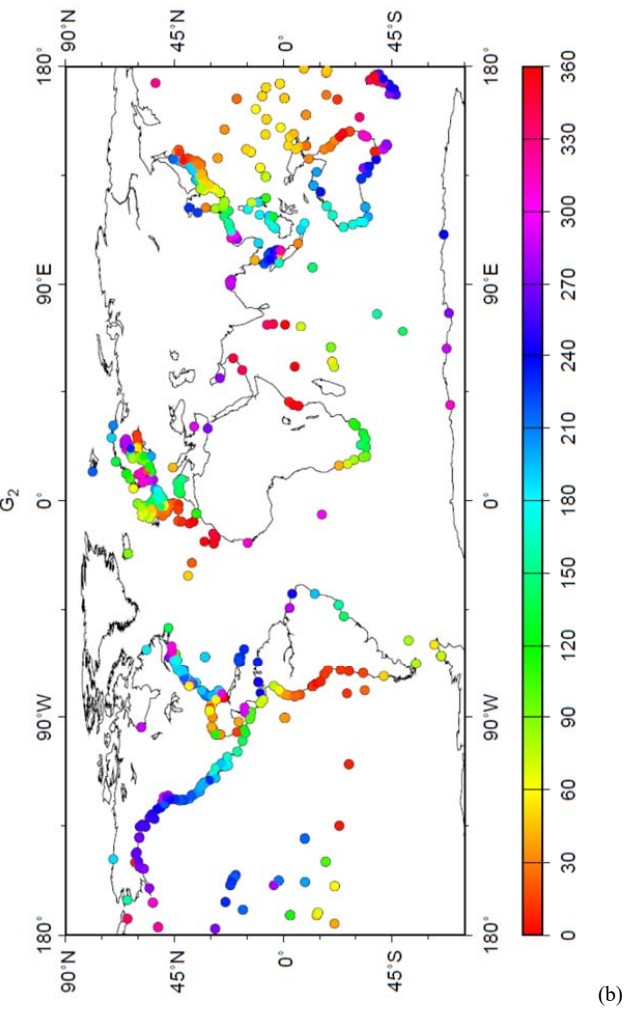

(b)

**Figure 5.** (a) Amplitude (mm) and (b) Greenwich phase lag (deg) of M1′ at stations in the GESLA-2 data set.

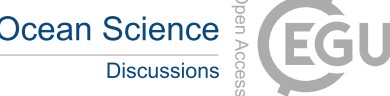


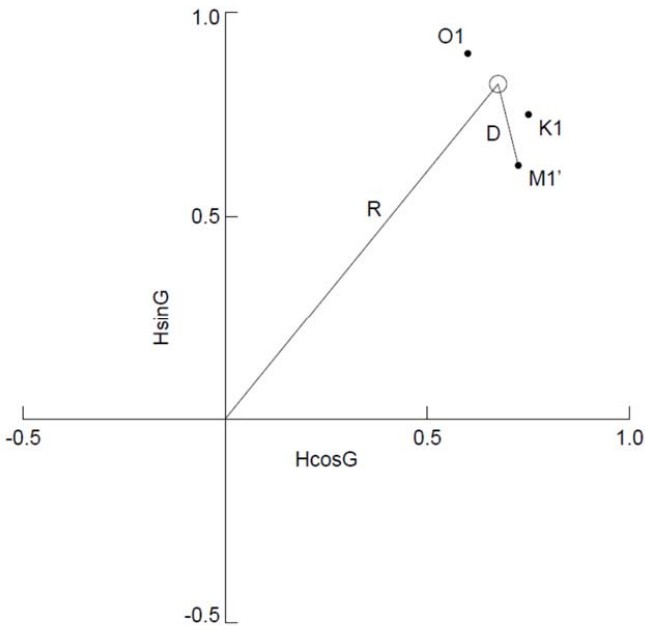

**Figure 6.** (a) An example of the positions of K1 and O1 in the complex plane, their mid-way position (open circle) and its amplitude (R), and the distance (D) between the measured M1′ and the mid-way point.

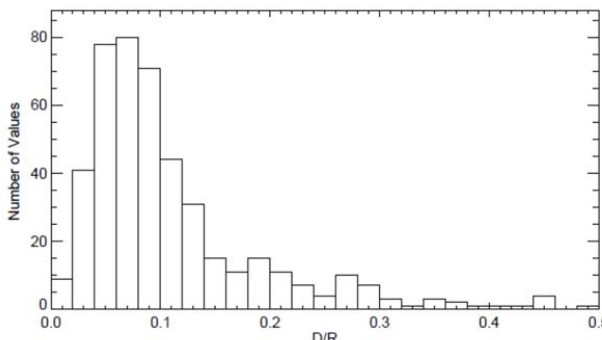

(b) A histogram of values of the ratio D/R obtained from records in GESLA-2 requiring the amplitude of K1 to be at least 10 cm.



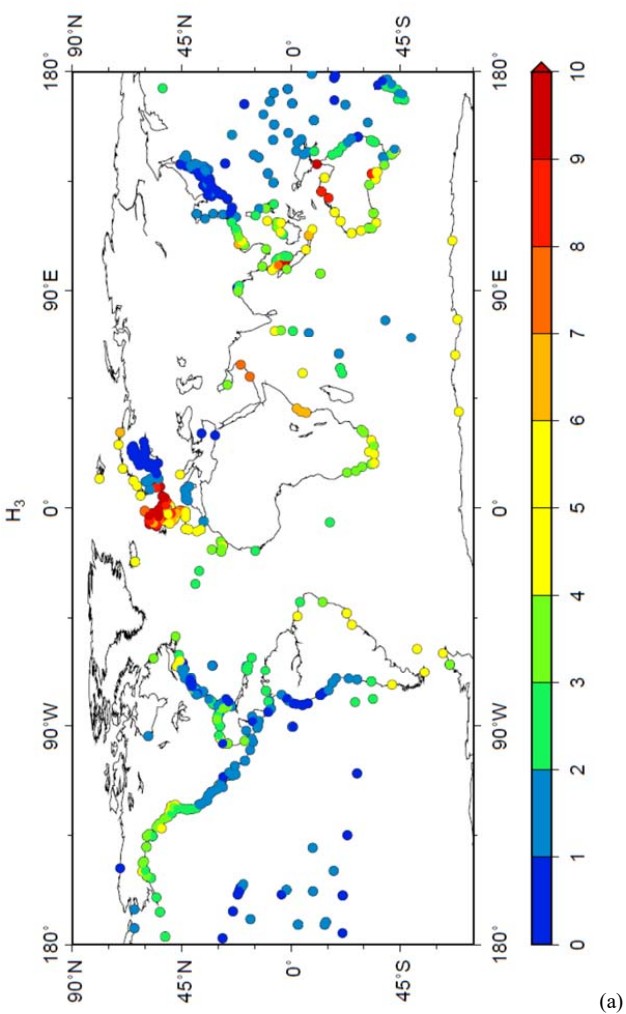

(a)



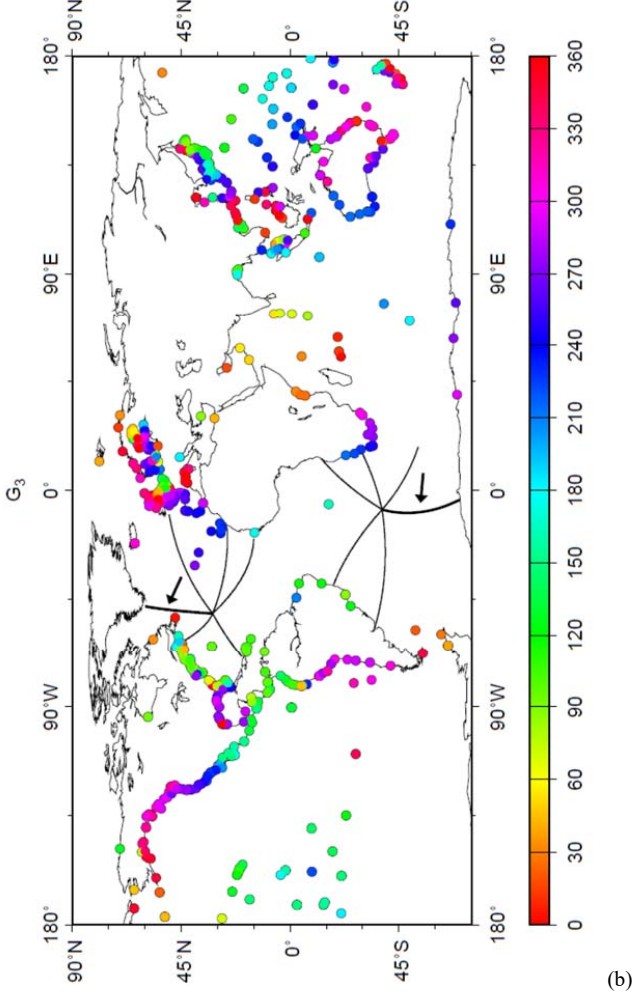

(b)

**Figure 7.** (a) Amplitude (mm) and (b) Greenwich phase lag (deg) of M1 at stations in the GESLA-2 data set. (b) includes the co-tidal lines from the Platzman synthesis for M1 shown in Figure 1.



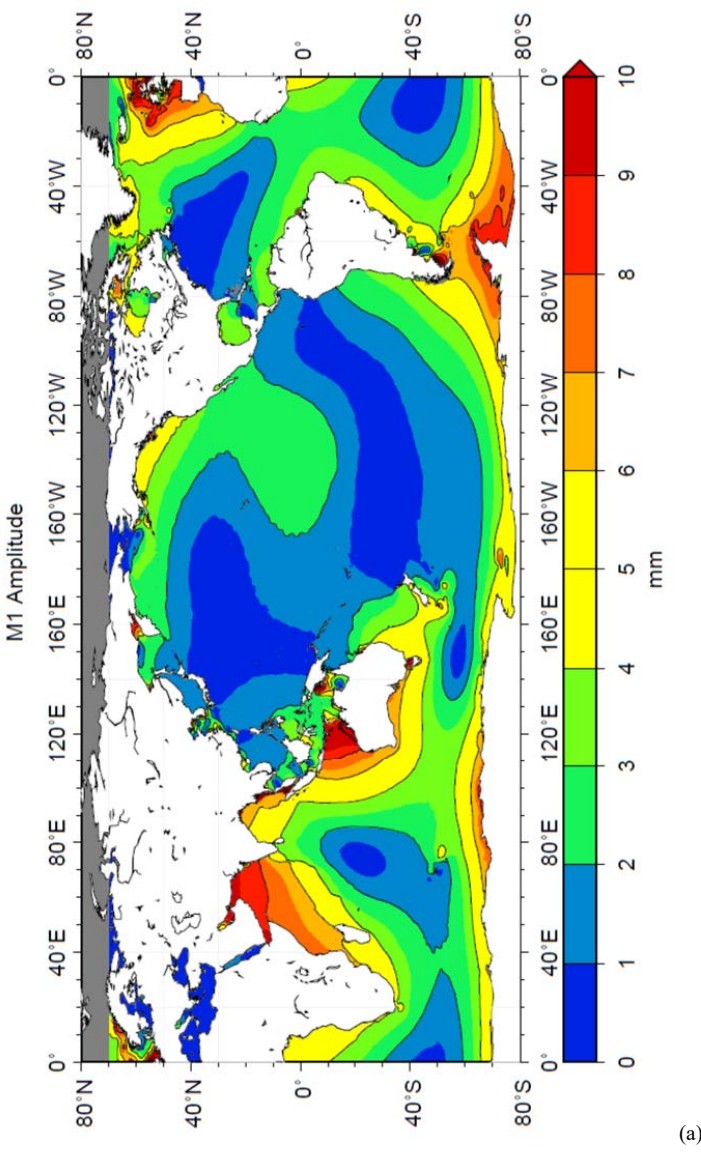

(a)

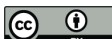



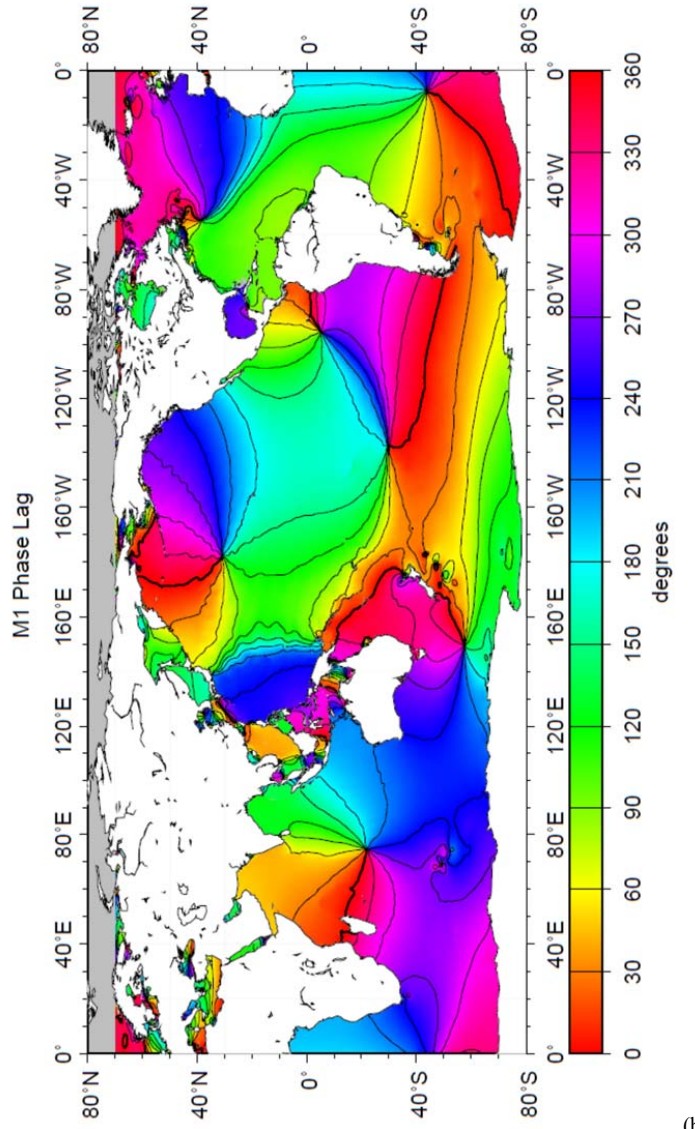

(b)

**Figure 8.** (a) Amplitudes and (b) Greenwich phase lags of M1 from the numerical model.