# Peer review of "The global distribution of the M1 ocean tide"

_Ocean Science, 2018_

## Short Comment (SC1) · 24 Jan 2019

Hi Phil, Thank-you for writing this up and the talk you gave on it last year. I'm sure it'll get sent to review a bit further away, but in the meantime here's a few points I've picked up on.

The distinction between M1 and M1' is quite hard to keep track of, especially as existing software conflates them, and M1' is the degree-2 constituent usually labelled M1. Is there a clearer notation you could adopt, eg d2-M1 for M1' and d3-M1 for your M1? It might also help to define this up front in case we end up looking at other degree-3 tides in the future. Also M1' is used in the abstract before definition.

Confusion with NO1 is also likely, especially for users of the Foreman-derived codes (including T_tide and U_tide). In these codes only NO1 is named in the standard constituent list, with the same frequency as line 7 from Table 1. (In contrast, only M1

is named in the NOCtide or TIRA list, with the same frequency as line 4). Could you clarify the difference?

Also, though P&W 2014 does have an explanation of the degree-2 & degree-3 polynomials, it's not easy to find unless you know what you are looking for. A brief explanation here would be useful.

The abstract could include an estimate of the maximum amplitude. Oh yes, and there's no scales on some of the amplitude maps (I was quite disappointed when I realised it was mm rather than cm!)

p7 line 21: Presumably large V could also arise from frequent tide-surge interaction, which may be a contributing factor in the North Sea? Figure (5c) doesn't exist, should it be Supp. Fig. 3c?

In Table 1, frequency is given as degrees/hour not cycles/hour.

Are there other significant degree-3 tides that you know about, or is M1 a lot bigger than the rest? What led you to pick up on M1?

Thanks, Jo

---

## Referee Comment (RC1) · Anonymous Referee #1 · 6 Feb 2019

The small degree-3 M1 ocean tide has been previously studied by analyzing data from regional (mostly North Atlantic) tide gauge networks and by synthesis of normal modes in the world ocean. The present work extends the tidal analysis of M1 and the nearby degree-2 M1' tide to the global ocean based on over 800 sea level records and a special parameterization of tidal constants and nodal factors in the frame of a least-squares fit. The distribution of M1 (and M1') in the various ocean basins is characterized using scatter plots of tidal constants and a numerically derived M1 solution.

This is a neat paper and indeed the first time that the ocean response to the degree-3 component of the tidal potential has been examined globally. The study is also a timely scientific contribution in light of the future SWOT (Surface Water and Ocean Topography) wide-swath mission, which will map shelf and coastal tides at high spatial resolution and with a target error of 1 cm. Having exactly this magnitude in several places, M1 will no longer be considered "esoteric"; instead, many groups will consult

the present paper for further detail on degree-3 diurnal tides and the cluster of degree-2 terms in the same spectral band (M1'). Although the numerical tidal modeling in the last part of the paper could be more sophisticated, it suffices the needs here. I have only a few minor corrections that should be addressed during the revision process:

– I found the second part of the Introduction (after the author sets out the objective) a bit jumpy and incoherent. References to the Cartwright and Platzman papers alongside re-prints of their figures are used to elucidate the main characteristics of the M1 tide. This section could do with some streamlining, e.g., by gluing the individual paragraphs together. Also, lines 13-14 describe symmetry properties of the degree-2 and degree-3 diurnal tidal potential, which are nicely illustrated in Fig. 2 of Ray (2001). It would make sense to aid a reader's imagination and directly refer him/her to Ray's figure.

– The M1' tide is introduced without any additional explanation in the abstract.

– The section on the prior tuning of the numerical model for leading degree-2 constituents (M2, K1) lacks some quantitative measures. Which values of horizontal eddy viscosity and bottom drag coefficient were used? Does the model run with a quadratic term for bottom friction or a simple linear parameterization? Instead of stating that the obtained maps for M2 and K1 are acceptable, include a comparison to data-constrained solutions in terms of RMS values and explained variances.

– Something that can be added to the comparison of the model-based M1 chart to the tide gauge estimates: similar to the author's solution for K1 (SI Fig. 1c), the model appears to overestimate amplitudes, particularly in the Southern Ocean/Indian Ocean. I have the strong suspicion that this results from treating Antarctic ice shelves as fully grounded and excluded sub-shelf cavities from the model domain. Check out the paper by Wilmes and Green (2014):

Wilmes, S.-B., and Green, J.A.M. (2014), The evolution of tides and tidal dissipation over the last 21,000 years, JGR Oceans, doi:10.1002/2013JC009605.

– page 5, line 2: "the tidal software" – is this the NOC software introduced in Section 2? The first paragraphs of Section 3 were not specific in this regard.

– The tidal potential used for forcing the tidal model is specified on line 22, page 10. I wonder if a bit more information on the amplitude is required at this place. Is this the amplitude of the equilibrium tidal potential listed in Table 1 of Ray (2001)? If yes, there might a slight mismatch in numbers (1.27 mm in Ray's paper vs. 1.2 mm here). The text also mentions a factor of 0.80 to account for the effect of elastic body tides (see below), but it is not clear whether or not the quoted formula includes this correction.

– Getting picky, yes, but using "Doodson number" in the second column of Table 2 is not fully correct. What is shown are actually the integer multipliers for the 6 Doodson variables that define the argument of the tidal term.

– Wahr (1981) is cited as Wahr (1991) in the both the main text and the supplement. I also recommend a slight re-formulation of the last sentence at the bottom of page 3: "... a special value of 0.74 for K1 to allow for resonant perturbations of body tide Love numbers close to the diurnal eigenfrequency of the Earth's fluid core."

– Finally, it always brings a smile on my face when a single author of a paper uses "we" in the active voice (such as in the abstract here). There can be different takes on it, but I to wonder with whom did he/she write the paper ...

---

## Referee Comment (RC2) · Ivan Haigh (Referee) · 5 Mar 2019

This paper used a global database of tide gauge records (GESLA) to examine, for the first time, the global distribution of the M1 tide. Overall, I found the paper to be very well written. The work will be of interest to tidal scientists. It is carried out robustly and clearly worthy of publication. My comments are very minor, as follows.

First, the final two paragraphs of the introduction seemed a bit out of place. I would suggest moving the sentence describing the overall aim of the paper to the end of the introduction sentence, and integrating the 2 last paragraphs into the introduction text a bit more coherently.

Second, in the data and methods section I would like a few more details on the tidal analysis. Exactly what software was used and maybe a weblink to this. I assume it is the NOC software described earlier.

[Figure]

Third, the paper needs more detailed on the numerical model used. At present this is described quite vaguely. It would be good to have a description of precisely what domain the grid covers, the co-ordinate system, how exactly the model is forced. I assume tidal potential. It might also be worth moving the description of the model to the data and methods section (3) and just describe the results in Section 4.3.

Fourth, the author uses the term 'we' which I find to be a bit strange given this is a single author paper.

Fifth, the author refers to Cartwrights findings, or Platzam, but it would be good to include a year after each reference, for those not so familiar with this literature.

Overall, I commend the author on a great paper.

---

## Author Comment (AC1) · 18 Mar 2019

Jo – thanks for the useful comments. Some replies below. Phil

*Hi Phil, Thank-you for writing this up and the talk you gave on it last year. I'm sure it'll get sent to review a bit further away, but in the meantime here's a few points I've picked up on.*

Thanks again.

*The distinction between M1 and M1' is quite hard to keep track of, especially as existing software conflates them, and M1' is the degree-2 constituent usually labelled M1. Is there a clearer notation you could adopt, eg d2-M1 for M1' and d3-M1 for your M1? It might also help to define this up front in case we end up looking at other degree-3 tides in the future. Also M1' is used in the abstract before definition.*

I agree that the different historical notations are confusing – you have to read all the papers and decide what each person was actually meaning by the different names (see my Footnote 2). There was an international attempt to standardise the names of tidal lines by the IHO Tides Committee (IHO, 2006) but I am not sure if everyone follows that. I don't think in the present case that the situation would be improved by inventing yet another set of notations. In my case, I was attempting to build on the previous work of Doodson and Cartwright, and so I adopted the Doodson-type definition of M1´ as I explained in the footnote. I have edited the abstract to define M1´ as suggested.

*Confusion with NO1 is also likely, especially for users of the Foreman-derived codes (including T_tide and U_tide). In these codes only NO1 is named in the standard constituent list, with the same frequency as line 7 from Table 1. (In contrast, only M1 is named in the NOCtide or TIRA list, with the same frequency as line 4). Could you clarify the difference?*

All tidal analysis software comes with its own history, and in the NOC case that was very much influenced by Doodson, hence the definition of M1´ as a combination of four lines as I explained, which is then confusingly called M1 in the software. Now, by far the largest of those four in the tidal potential (Table 1) is line (7), so I can see why in a different software the authors might chose to fit to that alone, as you say happens in T_tide and U_tide, rather than the combination of the four. As you know, in any routine tidal analysis one has a limited number of constituents to fit to and what is included in that list depends on the local research experience. I think NOAA software also fits just to line 7 (see Bruce Parker's NOAA tidal analysis manual) and in fact that was what Cartwright (1975) did, in his case confusingly giving the name M1´ to line (7) alone – see my footnote 2.

As for 'NO1', beware again of the usage of the name. These other packages (and also IHO, 2006) could well give the name 'NO1' to the fitted harmonic with the frequency of line 7. The packages will of course determine the amplitude and phase of the tide with that frequency that results in the ocean directly from line 7 in the tidal potential, plus any interaction of N2 and O1. In the present paper, we (i.e. in the quotes from the work of Richard Ray) refer to 'NO1' as just the interaction part. I hope that is clear from the text.

*Also, though P&W 2014 does have an explanation of the degree-2 & degree-3 polynomials, it's not easy to find unless you know what you are looking for. A brief explanation here would be useful.*

I do not want to go more into text book mode in what is a research paper. The degree-3 components of the tide come from the 3rd degree Legendre polynomial part of the tidal potential, which are described adequately in outline at least in Agnew (2007), and more briefly in Pugh (1984) and Pugh and Woodworth (2014). There is a cartoon of the degree-3 component of the tidal potential in Cartwright (1975). It is otherwise necessary to get to grips with Cartwright and Tayler (1971).

*The abstract could include an estimate of the maximum amplitude.*

Done. Thanks.

*Oh yes, and there's no scales on some of the amplitude maps (I was quite disappointed when I realised it was mm rather than cm!)*

Yes, things are very small (several mm usually or about 1 cm around the North Sea). There were units given on all the figures in the paper, either on the figures themselves or in the captions. I have remade figures 5 and 7 to ensure the units are now on all the figures.

*p7 line 21: Presumably large V could also arise from frequent tide-surge interaction, which may be a contributing factor in the North Sea?*

An interesting suggestion. I daresay if the mis-match between the left and right-hand sides of Equation 2 produces large V due to the complexities of tidal interaction (like NO1), then tide-surge interaction might also play a part, although it is hard to see how off-hand. I guess there is a need for a modelling study.

*Figure (5c) doesn't exist, should it be Supp. Fig. 3c?*

Thanks. Fixed.

*In Table 1, frequency is given as degrees/hour not cycles/hour.*

Thanks. Fixed.

*Are there other significant degree-3 tides that you know about, or is M1 a lot bigger than the rest? What led you to pick up on M1?*

M3 can be locally important in some shelf areas (e.g. see Huthnance, Deep-Sea Res, 1980). Otherwise, see the tables in Cartwright and Tayler for the relative importance of each line – most of the other degree-3 terms are very small, although they can cause complications by overlapping with degree-2 lines with similar frequencies, and of course familiar constituents such as N2 and L2 have perigean variations which are degree-3. I was interested in M1 myself because it was unfinished business from Cartwright et al. (1988).

---

## Author Comment (AC2) · 18 Mar 2019

Many thanks for these comments.

*This paper used a global database of tide gauge records (GESLA) to examine, for the first time, the global distribution of the M1 tide. Overall, I found the paper to be very well written. The work will be of interest to tidal scientists. It is carried out robustly and clearly worthy of publication. My comments are very minor, as follows.*

*First, the final two paragraphs of the introduction seemed a bit out of place. I would suggest moving the sentence describing the overall aim of the paper to the end of the introduction sentence, and integrating the 2 last paragraphs into the introduction text a bit more coherently.*

I see what you mean. I have moved the final three paragraphs around and reworded things slightly, so as to end the Introduction with the overall aim.

*Second, in the data and methods section I would like a few more details on the tidal analysis. Exactly what software was used and maybe a weblink to this. I assume it is the NOC software described earlier.*

Yes, it is the NOC software. I have made that clear now.

*Third, the paper needs more detailed on the numerical model used. At present this is described quite vaguely. It would be good to have a description of precisely what domain the grid covers, the co-ordinate system, how exactly the model is forced. I assume tidal potential. It might also be worth moving the description of the model to the data and methods section (3) and just describe the results in Section 4.3.*

I have added a few more words to the second paragraph of section 4.3. The second and third paragraphs already refer to the things the reviewer mentions above i.e. the model is global, ¼ degree grid, and forced only by the tidal potential. I have added now that it is a finite-difference model and mentioned once again that it is driven only by the tidal potential. There are full details in the Supplementary Material which the reader is pointed to in the text. That now has more model details that Reviewer 1 requested and that I should have provided already. I did not want to include 3-4 pages of model description and validation in the main part of the paper.

*Fourth, the author uses the term 'we' which I find to be a bit strange given this is a single author paper.*

Thanks. I agree. I did a search on 'we' and reworded things.

*Fifth, the author refers to Cartwrights findings, or Platzman, but it would be good to include a year after each reference, for those not so familiar with this literature.*

I know what you mean. I have included the year now in 4 or 5 more places. But sometimes it seems clumsy to refer to 'Cartwright (1975, 1976) and Cartwright et al. (1988)' when the reader by then will know what is meant by 'the three Cartwright papers' so I have left it as that in a couple of places.

*Overall, I commend the author on a great paper.*

Many thanks again.

---

## Author Comment (AC3) · 18 Mar 2019

Many thanks for the kind remarks about the paper. I have included below only the questions from your report.

- I found the second part of the Introduction (after the author sets out the objective) a bit jumpy and incoherent. References to the Cartwright and Platzman papers alongside re-prints of their figures are used to elucidate the main characteristics of the M1 tide. This section could do with some streamlining, e.g., by gluing the individual paragraphs together. Also, lines 13-14 describe symmetry properties of the degree-2 and degree-3 diurnal tidal potential, which are nicely illustrated in Fig. 2 of Ray (2001). It would make sense to aid a reader's imagination and directly refer him/her to Ray's figure.

Reviewer 2 had a similar comment concerning the Introduction. I have swapped around the final paragraphs in that and reworded things slightly so the Introduction should now read more logically. I have given a mention of Ray's Figure 2 as suggested.

- The M1' tide is introduced without any additional explanation in the abstract.

Jo Williams and Reviewer 2 also commented on this. It is defined now.

- The section on the prior tuning of the numerical model for leading degree-2 constituents (M2, K1) lacks some quantitative measures. Which values of horizontal eddy viscosity and bottom drag coefficient were used? Does the model run with a quadratic term for bottom friction or a simple linear parameterization? Instead of stating that the obtained maps for M2 and K1 are acceptable, include a comparison to data-constrained solutions in terms of RMS values and explained variances.

I should have given more details. Model construction, as a shelf model, is described in Flather (Trieste Summer School, 1988). It was run with a 15-second time step and saved sea levels at each grid point every hour (240 steps). Five days spin-up was followed by 14 days to give the values for tidal analysis. It used quadratic bottom friction, and after some experiments, a coefficient of 0.004 was chosen (0.0025 is more normal in 2-D models, see Heaps, 1978 or Pugh and Woodworth, 2014). The horizontal eddy viscosity (A) was depth-dependent with a coefficient AH of 15.0 m/sec (i.e. A=AH\*Depth in metres). Values of A are usually taken to be 100-1000 m2/sec in shelf models (Heaps, 1978), so the value of A used here is much larger than that in deep water but that provides the required 'glue-like' ocean (but with the wrong physics, see the Supplementary Material). I have added this information to the Supplementary Material. References are:

Flather, R.A. 1988. Storm surge modelling. Lecture notes from Ocean Waves and Tides Course, International Centre for Theoretical Physics, Trieste, Italy, sponsored by Proudman Oceanographic Laboratory, September 26 to October 28, 1988. Unpublished document. A pdf copy may be obtained from the present author.

Heaps, N.S. 1978. Linearized vertically-integrated equations for residual circulation in coastal seas. Deutsche Hydrografische Zeitschrift, 31, 147-169, doi:10.1007/BF02224467.

I assume the last sentence of this comment is suggesting a quantitative comparison of the present model in Supplementary Figure 1(c,d) to a state-of-the-art tide model such as that in Supplementary Figure 1(a,b). I did not do that simply because I knew that any quantitative comparison would not be much use. I am very aware that the model used here is crude by modern standards (and even by the standards of decades ago e.g. Accad and Peckeris, 1978), as I explained in the Supplementary Material, but in this case I considered it adequate for the job in hand.

- Something that can be added to the comparison of the model-based M1 chart to the tide gauge estimates: similar to the author's solution for K1 (SI Fig. 1c), the model appears to overestimate amplitudes, particularly in the Southern Ocean/Indian Ocean. I have the strong suspicion that this results from treating Antarctic ice shelves as fully grounded and excluded sub-shelf cavities from the model domain. Check out the paper by Wilmes and Green (2014): Wilmes, S.-B., and Green, J.A.M. (2014), The evolution of tides and tidal dissipation over the last 21,000 years, JGR Oceans, doi:10.1002/2013JC009605.

There is indeed a likely problem with handling Antarctic ice shelves in the model, together with the general uncertainties to do with Antarctic coastlines and bathymetry. I have inserted a mention of Wilmes and Green (2014) as an example of other authors having had similar difficulties, although I noticed that they had more problems with semidiurnals than diurnals.

- page 5, line 2: "the tidal software" – is this the NOC software introduced in Section 2? The first paragraphs of Section 3 were not specific in this regard.

Reviewer 2 also asked about this. I have made it clear now that I used the NOC software.

- The tidal potential used for forcing the tidal model is specified on line 22, page 10. I wonder if a bit more information on the amplitude is required at this place. Is this the amplitude of the equilibrium tidal potential listed in Table 1 of Ray (2001)? If yes, there might a slight mismatch in numbers (1.27 mm in Ray's paper vs. 1.2 mm here). The text also mentions a factor of 0.80 to account for the effect of elastic body tides (see below), but it is not clear whether or not the quoted formula includes this correction.

No, the two numbers are not the same. The 1.268 mm in Table 1 of Ray (2001) is the M1 equilibrium tide amplitude at Newlyn. If you multiply the 3.99 mm of the potential from Cartwright and Tayler by the Y31 spherical harmonic at Newlyn latitude (approx. 50.1) and then multiply by the degree-3 diminishing factor (approx. 0.8) then you will get to the 1.268 mm.

In my case the 1.2 mm (I apologise that was a mistake and it should have been rounded up to 1.3 mm) is the 3.99 mm of the potential times sqrt(7/(192\*pi)) \* 3 = 1.3 mm which then has to be multiplied by the geometrical terms as shown in the paper.

The quoted formula (the 1.3 mm) does not include the diminishing factor. I have reworded the sentence to make it clearer.

- Getting picky, yes, but using "Doodson number" in the second column of Table 2 is not fully correct. What is shown are actually the integer multipliers for the 6 Doodson variables that define the argument of the tidal term.

I see your point. I have changed the header of that column to 'Doodson numbers', not least because footnote 1 of the paper refers to the 'first two Doodson numbers'. But all six collectively can also be referred to as a 'Doodson number' (at least by me and see IHO, 2006 for example), or an 'Extended Doodson number' when the phase is included, although to be historically pedantic one should add the 5's as in Doodson's own tabulations. I have added a footnote to the table along these lines. Thanks for pointing this out.

- Wahr (1981) is cited as Wahr (1991) in the both the main text and the supplement. I also recommend a slight re-formulation of the last sentence at the bottom of page 3: "... a special value of 0.74 for K1

to allow for resonant perturbations of body tide Love numbers close to the diurnal eigenfrequency of the Earth's fluid core."

Wahr changed to 1981. I wanted to avoid mentioning unnecessary terms ('Love number' as well as 'diminishing factor'). I have changed the wording about the special value for K1 in order to mention Agnew (2018) which contains many useful references on this topic.

- Finally, it always brings a smile on my face when a single author of a paper uses "we" in the active voice (such as in the abstract here). There can be different takes on it, but I to wonder with whom did he/she write the paper ...

Point taken. Reviewer 2 also mentioned this. I made a search on 'we' and reworded things.

Many thanks again for these useful comments.

---

## Author Response (AR1)

National Oceanography Centre
NATURAL ENVIRONMENT RESEARCH COUNCIL

Joseph Proudman Building
6 Brownlow Street
Liverpool
Merseyside, L3 5DA
United Kingdom

Tel:  +44 (0) 151 795 4800
Fax: +44 (0) 151 795 4801

www.noc.ac.uk

Editor, Ocean Science

20 March 2019

Dear Sir,

Paper Submitted to Ocean Science

Many thanks for the reviews of my paper submitted to Ocean Science Discussions entitled "The global distribution of the M1 ocean tide". I have responded to the comments of the reviewers, and also to another interactive comment, via the web system.

The paper has since been edited following these comments, and several new edits of my own have been made. Figures 5, 7 and 8 have been remade. I have uploaded the new version and below you will find a comparison using Word of the edits made.

Could I raise an issue again which occurred with a previous paper submitted to this journal last year? That was to do with the fact that I am retired and have no access to institutional funds. Copernicus was kind enough to waive the charges for that previous paper. I daresay it will be harder to push my luck to the same extent again. Nevertheless, if the present paper is accepted for Ocean Science, I would be very grateful if the charges could be set as low as possible. This special issue is the obvious journal for me to have submitted this paper to.

If there are problems, I can be contacted at plw@noc.ac.uk.

Many thanks for your help with this.

Yours sincerely

Philip Woodworth

[revised manuscript text omitted]

---

## Author Response (AR2)

There are no more responses to reviewers to make than I made in the replies to the interactive comments on the web site at the end of the OSD stage.